# Biomass Smoke Exposure and Atopy among Young Children in the Western Highlands of Guatemala: A Prospective Cohort Study

**DOI:** 10.3390/ijerph192114064

**Published:** 2022-10-28

**Authors:** Wenxin Lu, Laura Ann Wang, Jennifer Mann, Alisa Jenny, Carolina Romero, Andrea Kuster, Eduardo Canuz, Ajay Pillarisetti, Kirk R. Smith, John Balmes, Lisa Thompson

**Affiliations:** 1School of Public Health, University of California, Berkeley, CA 94720, USA; 2Department of Pediatrics, University of Colorado, Aurora, CO 80045, USA; 3Institute for Global Health Sciences, University of California, San Francisco, CA 94158, USA; 4Centro de Estudios en Salud, Universidad del Valle de Guatemala, Guatemala City 01015, Guatemala; 5School of Nursing, University of California, San Francisco, CA 94158, USA; 6Nell Hodgson Woodruff School of Nursing, Emory University, Atlanta, GA 30322, USA

**Keywords:** CRECER study, biomass smoke, household air pollution, child health, allergy, asthma, rhinitis, eczema

## Abstract

Women and children in rural regions of low-income countries are exposed to high levels of household air pollution (HAP) as they traditionally tend to household chores such as cooking with biomass fuels. Early life exposure to air pollution is associated with aeroallergen sensitization and developing allergic diseases at older ages. This prospective cohort study assigned HAP-reducing chimney stoves to 557 households in rural Guatemala at different ages of the study children. The children’s air pollution exposure was measured using personal CO diffusion tubes. Allergic outcomes at 4–5 years old were assessed using skin prick tests and International Study of Asthma and Allergies in Childhood (ISAAC)-based questionnaires. Children assigned to improved stoves before 6 months old had the lowest HAP exposure compared to the other groups. Longer exposure to the unimproved stoves was associated with higher risks of maternal-reported allergic asthma (OR = 2.42, 95% CI: 1.11–5.48) and rhinitis symptoms (OR = 2.01, 95% CI: 1.13–3.58). No significant association was found for sensitization to common allergens such as dust mites and cockroaches based on skin prick tests. Reducing HAP by improving biomass burning conditions might be beneficial in preventing allergic diseases among children in rural low-income populations.

## 1. Introduction

Approximately 2.4 billion people worldwide rely on burning biomass fuels, including wood, crop residue, coal, and animal dung for household cooking and heating [1]. These fuels are often burned indoors in simple stoves in often poorly ventilated rooms, emitting substantial amounts of pollutants including particulate matter, carbon monoxide, and polycyclic aromatic hydrocarbons, all components of household air pollution (HAP). It is estimated that HAP from biomass fuel is responsible for more than 1.8 million premature deaths and 60.9 million disability-adjusted life-years annually, with the greatest burden experienced in low- and middle-income countries [2]. Women and children have the highest levels of exposure to HAP since they traditionally tend to daily household chores, including cooking and spending time in kitchens [3]. Exposure to biomass smoke during childhood has been shown to impact lung function and increase risk of acute lower respiratory infections [4,5]. However, less is known about the rates of allergic sensitization and subsequent risk of atopy, which is the tendency to develop allergic diseases such as allergic rhinitis, eczema, and allergic asthma, among children who are exposed to biomass smoke early in childhood.

Prior studies focusing on outdoor air pollution suggest that exposure to traffic-related air pollution early in life may enhance the risk of aeroallergen sensitization and increase the risk of developing allergic diseases at older ages [6,7,8,9]. Diesel exhaust, in particular, has been found to enhance risk of aeroallergen sensitization and promote development of allergic asthma [10,11]. Environmental tobacco smoke exposure early in life has also been associated with the development of allergic sensitization in children [12,13]. Studies that investigated cooking-related biomass smoke exposure in children have also found it to be associated with elevated risks of asthma [14,15,16,17,18], rhinitis [18,19], and other respiratory illnesses [14,18].

There are, however, limited data regarding the relationship between HAP, aeroallergen sensitization, and atopy in pediatric populations. A cross-sectional study conducted in rural Bavaria in southern Germany in a cohort of school-aged children from 9 to 11 years old found that children living in households using coal or wood for heating were less atopic and less likely to be sensitized towards aeroallergens than children living in households with other forms of heating [20]. In a large study of Finnish children, there was also a negative association between woodstove heating and allergic rhinoconjunctivitis; however, this association disappeared after adjusting for other factors in the residential environment [21]. In another population of children in Spain, the use of a biomass energy source for cooking and indoor heating was not associated with atopic dermatitis [22]. Contrastingly, for children in rural Guatemala, a previous study suggested that the use of an open fire for cooking may be an important risk factor for asthma symptoms and severity later in life [23].

The primary objective of this study is to examine the relationship between early biomass smoke exposure and atopy among a cohort of children enrolled in a HAP-reducing chimney stove intervention trial among a population living in the western highlands of Guatemala. In the study community, women often carry their youngest child on their back during cooking, until the child is approximately 18 months old [3], exposing the newborn children to high levels of HAP. We hypothesize that the availability of a vented chimney stove would reduce the children’s HAP exposure compared to those who use open fires for cooking and would be associated with risks of allergic sensitization.

## 2. Materials and Methods

### 2.1. Study Design and Participants

Participating households and children were recruited from the Randomized Exposure Study of Pollution Indoors and Respiratory Effects (RESPIRE) cohort and its follow-up study, the Chronic Respiratory Effects of Early Childhood Exposure to Respirable PM (CRECER) study. Details of the RESPIRE and CRECER cohorts have been published elsewhere [4,24,25]. Briefly, 518 rural Guatemalan women with newborn children who cooked exclusively over an open fire were recruited for the RESPIRE study between October 2002 and December 2004. Households were randomized to either receive a chimney stove (*plancha*), which improves combustion and uses a chimney to vent emissions outdoors, or to continue to cook with their typical open fires until the end of the trial, when they also received the intervention *plancha* stove. 

CRECER, the follow up study, took place from 2006 to 2009 and revisited RESPIRE households and recruited 169 new households that (1) were from the same geographical region, (2) exclusively used open fires, (3) had one child (index study child) in the same age range as the RESPIRE study children (3–4 years old at the time of CRECER) and one infant less than 6 months old (proxy infant sibling). For equity purposes, these new households received a chimney stove at the end of the CRECER study when all exposure and outcome information had already been collected. 

All households in the study thus received a chimney stove at different time periods: RESPIRE intervention households (group 1) received the stove when the index children were less than 6 months old; RESPIRE control households (group 2) received the stove when the index children were approximately 18 months old; new CRECER households (group 3) received the stove when the index children were approximately 5 years old, and their proxy infant siblings were 18–24 months old. The grouping and study timeline are illustrated in Figure 1. 

### 2.2. Biomass Smoke Exposure

The *plancha* stoves provided in this study reduced the children’s biomass smoke exposure by improving combustion and venting cooking smoke outdoors. Although this may result in an increase in outdoor exposure, we would expect participants in households with *plancha* stoves to have lower overall biomass smoke exposure, because the total amount of smoke produced would not increase, and outdoor biomass smoke would affect children for shorter duration and lower intensity. We also hypothesized that group 1 index study children would have the lowest cumulative biomass smoke exposure because they were provided the *plancha* stoves earliest, followed by groups 2 and 3, respectively. To test these hypotheses, we measured carbon monoxide (CO) exposure for the study children.

Personal CO exposure was used as a proxy for personal biomass smoke exposure: CO has been shown to correlate well with fine particulate matter (PM 2.5) exposure in this population, in homes using open fires or chimney stoves [26,27,28]. Study participants wore small, passive CO diffusion tubes for 48 h every 3 months during RESPIRE and every 6 months during CRECER. Since group 3 index study children did not have CO measurements obtained when they were <18 months of age, the personal CO exposures of their younger infant siblings were used as a proxy for their early life exposures. Details on exposure assessment methodology, validation, and quality control and assurance have been extensively described elsewhere [29,30].

We combined data from different measurements, including the aforementioned 48-h samples, to estimate cumulative CO exposure. We used RESPIRE CO tube measurements to estimate cumulative exposure during the first 600 days of life for groups 1 and 2 and used CRECER infant sibling CO tube measurements to estimate cumulative exposure during the first 600 days of life for group 3. Cumulative CO exposure from 600 days old to first allergy questionnaire was estimated based on CRECER CO tube measurements for the study children. We conducted a sensitivity analysis using three alternative calculations for cumulative CO exposure, two of which did not use the 600 days cut point. Details of these calculations can be found in the Table A1.

### 2.3. Skin Prick Tests

Five rounds of skin prick tests (SPTs) were performed on 539 participants to determine allergic sensitization to six common indoor and outdoor aeroallergens (house dust mites *D. farina* and *D. pteronyssinus*, cockroach, dog, cat, and ragweed). During each round of SPT, a positive control (histamine) and a negative control (saline) were also performed on each child. A positive SPT result was indicated by a wheal measurement 3 mm or greater. SPT results were considered invalid if the histamine control was negative or the saline control was positive. Results from children who reported taking antihistamine or cold medications prior to testing were also excluded.

### 2.4. Allergic Outcome Questionnaires

Atopic symptoms were assessed via quarterly respiratory questionnaires (QRQs) completed by the study children’s mothers. The QRQs were conducted three times during CRECER to ascertain the occurrence of symptoms associated with asthma, allergic rhinitis, and eczema. The questions were developed based on the International Study of Asthma and Allergies in Childhood (ISAAC) questionnaire [31]. All QRQs were conducted by field workers fluent in the participating mothers’ primary language (Mam). Details of the questionnaire development and translation processes have been published elsewhere [32]. All questions were close ended (Yes/No/Don’t know) and began with “in the last three months”. A child’s final allergic outcome was recorded as positive if the mother reported him/her having positive symptoms in any of the three QRQ rounds.

### 2.5. Statistical Analysis

Logistic regression models were used to analyze the relationship between biomass smoke exposure and the risks of developing allergic outcomes. Primary statistical analysis used study group as a categorical exposure variable based on the length of having a chimney stove in the household. Odds ratios (ORs) and 95% confidence intervals (CIs) were reported. In this analysis, group 1 is the baseline level, groups 2 and 3 represent intermediate and highest levels of biomass smoke exposure, respectively. Secondary statistical analysis used the estimated cumulative CO exposure as a linear continuous exposure variable.

Age, sex, second-hand smoke exposure, the number of children in the family, kitchen structure (whether the kitchen was in a separate or the same building as the living area, and whether the kitchen was open or partitioned), child’s average weekly *temazcal* (wood-fueled sauna bath) use in minutes, number and species of pets and farm animals at home, maternal history of allergic outcomes, parental education, and socioeconomic status (SES) were collected from the CRECER baseline questionnaire, and were adjusted for in the logistic regression models. We did not adjust for race because the study population was homogenous, self-identifying as Mam indigenous.

## 3. Results

Among the 557 households participating in CRECER, 20 lacking valid SPT or QRQ results were excluded. For the two households with twins that both participated in the study, only the first child recorded was kept in the analysis to ensure independence among observations. Respiratory outcomes were available for 537 children, 188 from group 1, 192 from group 2 and 157 from group 3. Valid SPT results were available for 526 children, 184 from group 1, 187 from group 2 and 155 from group 3 (Figure 1). The quality and precision of the five rounds of SPTs during CRECER improved with additional training and more experience of the staff, indicated by the significantly lower number of invalid tests in the later rounds. As such, the results of the last valid round of SPT for each child was recorded as the child’s final allergic sensitization outcome. Among the 526 children with valid SPT results, results for 496 participants were taken from the 5th (last) round of SPT tests. 

The household-level and child-level demographic characteristics were similar among the three study groups (Table 1), especially between groups 1 and 2 that were randomized in the RESPIRE study. Study children were on average 3.6 years old at the first QRQ. Compared with groups 1 and 2, group 3 had higher proportions of maternal history of asthma, rhinitis and eczema, and slightly lower SES.

Table 2 and Figure 2 present summary statistics and box plots of the average and cumulative CO exposure levels for the 3 study groups, respectively. In both RESPIRE and CRECER, children in groups with chimney stoves had lower average CO exposures. Within groups, average CO exposure was lower when study children were older (during CRECER), which is consistent with the behavioral pattern that young children < 18 months old are often carried on their mothers’ backs during cooking, resulting in higher biomass smoke exposure. Among the three groups, group 2 has the highest cumulative CO exposures (6.21 ppm-year, SD 3.19 ppm-year), followed by group 3 (5.12 ppm-year, SD 3.07 ppm-year) and group 1 (4.16 ppm-year, SD 2.69 ppm-year). The differences in both average and cumulative CO levels among the study groups were statistically significant based on the one-way analysis of variance (ANOVA) test.

The proportions of positive SPT and QRQ outcomes differed by study groups (Table 3). Compared with groups 1 and 2, group 3 had higher prevalence of asthma-related wheezing and coughing, as well as allergic rhinitis symptoms. The most common outcome in the study population was allergic sensitization to cockroaches, affecting 37% of the study children, followed by asthma-related coughing (29%), and allergic rhinitis symptoms (25%). We excluded allergy to ragweed from the following statistical analyses because only two children were affected. 

Table 4 summarizes the logistic regression results of SPT and QRQ outcomes associated with study groups, where group 1 was treated as the reference group. After adjusting for age, sex, second-hand smoke exposure, *temazcal* use, kitchen structure, family history, animals owned, parental education and SES, group 3 had significantly higher odds of asthma-related wheezing symptoms (OR = 2.42, 95% CI: 1.11–5.48) and allergic rhinitis symptoms (OR = 2.01, 95% CI: 1.13–3.58). Allergy to dogs among group 2 was also significantly higher than group 1, but the extremely wide CIs flagged the lack of power due to the limited number of cases. Apart from dog allergy, there was no significant difference in other outcomes between groups 1 and 2. 

Table 5 presents the results for the secondary statistical analysis where the cumulative CO exposure summarized in Table 2 and Figure 2B was used as a linear continuous exposure. A 1 ppm-year higher cumulative CO exposure was associated with higher risks of developing allergic rhinitis symptoms (OR = 1.09, 95% CI: 1.02–1.18) and allergic conjunctivitis symptoms (OR = 1.11, 95% CI: 1.02–1.21), after adjusting for all covariates (Table 5). Sensitivity analyses using three other ways of cumulative CO calculations yielded consistent results: ORs for allergic rhinitis symptoms ranged from 1.06 (95% CI: 1.00–1.13) to 1.08 (95% CI: 1.00–1.17); OR for allergic conjunctivitis symptoms ranged from 1.07 (95% CI: 1.00–1.15) to 1.12 (95% CI: 1.03–1.22).

## 4. Discussion

This prospective cohort study followed more than 500 children in rural Guatemala over 7 years and examined associations between cooking-related biomass smoke exposure and childhood atopy outcomes. Children from households that received chimney stoves when the children were approximately 5 years old had higher risks of maternal-reported allergic asthma (OR = 2.42, 95% CI: 1.11–5.48) and rhinitis symptoms (OR = 2.01, 95% CI: 1.13–3.58) compared to children from households that received a chimney stove intervention within the first 6 months of life. A 1 ppm-year higher cumulative CO exposure and its related cumulative biomass smoke exposure was associated with 6–12% higher odds of maternal-reported allergic rhinitis and conjunctivitis symptoms. No significant association was found between biomass smoke exposure and eczema or skin prick test outcomes. Notably, the overall prevalence of sensitization to cockroach in this population of rural Guatemalan children was high (36–38%), which is similar to that reported for inner city children in the U.S [33]. 

A summary of main results from studies that looked at household biomass smoke exposure and allergic or respiratory outcomes is presented in Table A2. Our finding that higher biomass smoke exposure was associated with maternal-reported respiratory symptoms such as wheezing, sneezing, nasal congestion and rhinorrhea in their children is consistent with other studies that looked at exposure to biomass stove use and self-reported or clinically diagnosed respiratory diseases among children [15,23,34,35,36]. Prior studies that looked at exposure to cooking and heating-related HAP and atopy outcomes in children residing in high-income countries did not find significant associations after adjusting for lifestyle and socioeconomic factors [20,21,22]. In this study, we did not find a significant association between biomass smoke exposure and maternal-reported eczema symptoms. Additionally, there was no significant association between biomass smoke exposure and allergic sensitization as measured by SPT. Part of the reason might be that important windows for atopic sensitization such as prenatal exposures were not captured in the study [37,38,39]. The number of cases of allergic sensitization to dogs, cats, and ragweed were small among the study children (Table 3), resulting in compromised statistical power. The low number of dog and cat allergies might have been due to the high dog ownership (>80% for all groups) and medium cat ownership (~40%) in the study population (Table 1): living in proximity to animals is associated with lower sensitization to allergens among children [40,41]. 

No significant differences in allergic sensitization or symptoms were found between children in group 1 and group 2, among whom chimney stoves were installed around birth and around 18 months old, respectively. This might have been due to the gradual deterioration of chimney stoves during the 2-year gap between the RESPIRE and CRECER studies, during which group 1 might have been exposed to higher HAP than group 2 because of the older stoves. Another reason might be insufficient exposure reduction, which was also found in a previous analysis of the RESPIRE study: a larger reduction in mean CO exposure was associated with reduction in pneumonia risks, but the moderate difference in group mean CO levels between groups 1 and 2 was not enough to yield a statistically significant difference in pneumonia risk between the groups [24].

The high percentages of reported allergic symptoms and high prevalence of cockroach sensitization among the children in this study is contrary to the “hygiene hypothesis” or “microbial deprivation hypothesis” that early life exposure to microorganisms shapes the Th1 (T Helper 1 cells), Th2, and regulatory T cell responses and alters immune response patterns [42,43,44,45]. For instance, children exposed to enteric pathogens have higher resistance to allergic sensitization compared to those living in pathogen-free environments [46,47]. While the study population was exposed to abundant microorganisms, it is possible that exposure to HAP prenatally, in early life, or even in reduced amounts after the stove upgrade intervention, could promote a shift toward Th2 responses and thus increase risk for atopy. Previous studies have demonstrated that exposure to PM2.5 may increase the risk of asthma via airway inflammation, increase in oxidative stress, changes in immune signaling, and subsequent disruptions of airway epithelial cells and mucosal barrier function [48,49,50,51]. Studies on rhesus monkeys have also found that co-exposure to a pollutant that causes oxidative stress, ozone, and allergen altered airway structural development and increased risk of an asthma-like phenotype [52,53]. Another consideration is that the increased wheezing and rhinitis symptoms reported by their mothers among children exposed to higher HAP could also be due to the direct irritating effects of biomass smoke to the upper and lower airway epithelium rather than an underlying allergic mechanism [54,55,56].

During study design, we hypothesized that group 3 index study children would have the highest cumulative biomass smoke exposure because they were provided the upgraded chimney stoves the latest, thus were exposed to higher levels of biomass smoke for the longest period of time. We also expected groups 2 and 3 to have comparable levels of biomass smoke exposure during the RESPIRE study period because both groups did not have upgraded chimney stoves at this time. While assessing cumulative CO exposure for the index study children, we used the group 3 proxy infant siblings’ personal CO exposure during CRECER as a proxy for group 3 index study children’s personal CO exposure during the RESPIRE study period when the personal CO exposures of groups 1 and 2 were measured. This allowed us to account for the missing early life personal CO information due to the late recruitment of the group 3 households. This approximation assumes that newborn children raised in the same household by the same parent(s) will have similar activity patterns and thus similar levels of CO exposure. However, several sources of uncertainty may compromise the accuracy of this proxy measure. Firstly, secular differences between RESPIRE (2002–2004) and CRECER (2006–2009), such as different biomass fuels used, different CO diffusion tube batches, as well as the potential changes in household cooking conditions and ventilation, were not accounted for. Secondly, group 3 proxy infant siblings’ CO exposures were measured less frequently (every 6 months, during CRECER), compared to groups 1 and 2 index study children’s early life CO exposures (every 3 months, during RESPIRE), resulting in potential differential exposure misclassification. These uncertainties might be the reasons that the estimated CO exposure during the RESPIRE study period is much lower for group 3 compared to group 2 (Figure 2A), and the subsequent lower cumulative CO exposure for group 3 compared to group 2 (Figure 2B), which was different from our original hypothesis. If group 3 CO exposures were indeed underestimated, it would have caused a downward bias of our secondary analysis results (Table 5) because of the high number of cases in group 3, and the true associations would be higher than reported. 

This is the first cohort study that looked at SPT prevalence in a rural population of a low-income country. The strengths of this study include the quality of the SPTs, which was supported by positive and negative controls and multiple rounds of field worker training, and use of extensive questionnaires on household information, building structure, animal allergen exposures and SES to allow adequate control of potential confounding variables. The partial randomized controlled trial (RCT) design between groups 1 and 2 further reduced the possibility of residual confounding. The estimation of cumulative CO exposure was based on repeated personal measurements, which was of higher accuracy than commonly used ambient or static household air pollution monitors. 

An important limitation of the study was the self-reporting of allergic symptoms. Since the stove upgrade intervention could not be blinded, the mothers’ responses to questionnaires may have been subject to an upward response bias. In addition, the missing exposure information and stove deterioration during the 2-year gap between the RESPIRE and CRECER studies may have led to exposure misclassifications. The cumulative CO exposure estimation was also less accurate for group 3 because of the use of infant siblings as proxy for the early life exposure of the older children in this group, as well as the less frequent exposure monitoring during CRECER compared to RESPIRE, potentially resulting in differential exposure misclassification in the secondary analysis. 

## 5. Conclusions

In this two-stage prospective cohort study comparing children from households assigned to stove improvement interventions at different stages, elevated risks of experiencing allergic symptoms at around 4 to 5 years old were found to be associated with higher cooking-related biomass smoke exposure. Compared with groups that received chimney stoves before the children reached 18 months old, children who did not receive improved stoves until 5 years old reported higher risks of allergic asthma and rhinitis symptoms. Future household air pollution intervention studies should consider more objective measurements of allergic outcomes, characterization and evaluation of region-specific allergens, larger sample sizes, and cleaner stoves and fuels to provide more insight on this topic. 

## Figures and Tables

**Figure 1 ijerph-19-14064-f001:**
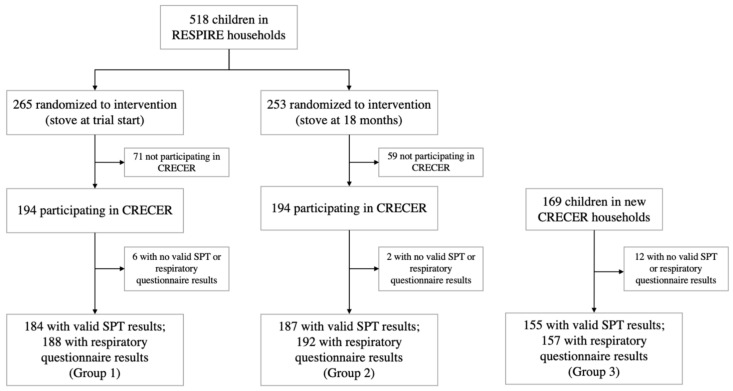
Study diagram and timeline for the Randomized Exposure Study of Pollution Indoors and Respiratory Effects (RESPIRE) and the Chronic Respiratory Effects of Early Childhood Exposure to Respirable PM cohort study (CRECER).

**Figure 2 ijerph-19-14064-f002:**
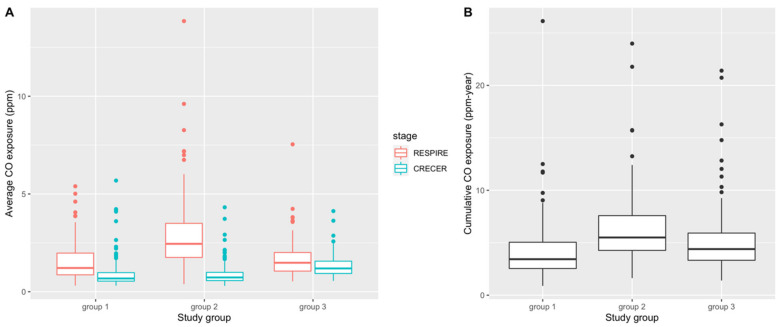
Distribution of average CO exposure levels (ppm) (**A**) and cumulative CO exposure (ppm-year) (**B**) during RESPIRE and CRECER study stages before first allergy questionnaire. Group 1 = <6 months; Group 2 = ~18 months and Group 3 = ~57 months).

**Table 1 ijerph-19-14064-t001:** Household-level and child-level demographic characteristics by study group.

	Group 1	Group 2	Group 3
*Plancha* installation age	(<6 months)	(~18 months)	(~57 months)
Number of children	n = 188	n = 192	n = 157
**Sex**			
Female	99 (52.7%)	101 (52.6%)	82 (52.2%)
**Age at first allergy questionnaire**			
Mean (SD)	3.57 (0.487)	3.55 (0.561)	3.57 (0.756)
**Maternal race**			
Indigenous (Mam)	185 (98.4%)	190 (99.0%)	156 (99.4%)
**Paternal race**			
Indigenous (Mam)	177 (94.1%)	186 (96.9%)	156 (99.4%)
**Number of children in household**			
Mean (SD)	5.01 (2.37)	4.98 (2.41)	4.69 (2.42)
**Second-hand smoke**			
Smoker in household	14 (7.4%)	22 (11.5%)	16 (10.2%)
**Kitchen type**			
Single structure, open kitchen	31 (16.5%)	31 (16.1%)	17 (10.8%)
Single structure, partitioned kitchen	12 (6.4%)	8 (4.2%)	7 (4.5%)
Separate structure, open kitchen	19 (10.1%)	18 (9.4%)	30 (19.1%)
Separate structure, partitioned kitchen	126 (67.0%)	135 (70.3%)	103 (65.6%)
**Temazcal (native steam bath) use**			
Temascal at home	147 (78.2%)	147 (76.6%)	117 (74.5%)
Average weekly temascal use time of the child (minutes): mean (SD)	26.3 (17.3)	26.8 (19.9)	27.6 (17.0)
**Pets and farm animals at home**			
Cattle: mean (SD)	0.62 (0.91)	0.55 (0.78)	0.59 (0.85)
Sheep: mean (SD)	1.62 (3.69)	1.25 (2.20)	1.58 (3.21)
Horses, mules, donkeys: mean (SD)	0.27 (0.53)	0.34 (0.58)	0.31 (0.60)
Pigs: mean (SD)	1.82 (1.86)	1.54 (1.50)	1.52 (1.61)
Poultry: (y/n)	180 (95.7%)	179 (93.2%)	152 (96.8%)
Dogs: (y/n)	170 (90.4%)	177 (92.2%)	127 (80.9%)
Cats: (y/n)	89 (47.3%)	81 (42.2%)	61 (38.9%)
Pigeons: (y/n)	15 (8.0%)	13 (6.8%)	16 (10.2%)
Other animals: (y/n)	21 (11.2%)	15 (7.8%)	18 (11.5%)
**Family history of allergic outcomes**			
Maternal asthma	30 (16.0%)	33 (17.2%)	52 (33.1%)
Maternal rhinitis	20 (10.6%)	26 (13.5%)	42 (26.8%)
Maternal eczema	24 (12.8%)	28 (14.6%)	34 (21.7%)
**Maternal education**			
None	64 (34.0%)	73 (38.0%)	46 (29.3%)
Primary school	117 (62.2%)	116 (60.4%)	109 (69.4%)
Middle school or higher	7 (3.7%)	3 (1.6%)	2 (1.3%)
**Paternal education**			
None	22 (11.7%)	26 (13.5%)	19 (12.1%)
Primary school	133 (70.7%)	130 (67.7%)	110 (70.1%)
Middle school or higher	24 (12.8%)	28 (14.6%)	27 (17.2%)
Unknown	9 (4.8%)	8 (4.2%)	1 (0.6%)
**Socioeconomics (SES)**			
Land-owning	169 (89.9%)	172 (89.6%)	125 (79.6%)
Home-owning	158 (84.0%)	167 (87.0%)	120 (76.4%)
Number of major assets *: mean (SD)	1.68 (1.01)	1.71 (1.13)	1.59 (1.08)

* Major assets include radios, televisions, refrigerators, bikes, motorcycles, automobiles, and cellphones.

**Table 2 ijerph-19-14064-t002:** Carbon monoxide (CO) exposure levels during RESPIRE and CRECER, and cumulative CO exposure before allergy questionnaire by study group.

	Group 1	Group 2	Group 3	
*Plancha* installation age	(<6 months)	(~18 months)	(~57 months)	One-way ANOVA *p*-value
	mean (SD)	mean (SD)	mean (SD)	
	median [Min, Max]	median [Min, Max]	median [Min, Max]	
CO exposure during RESPIRE (ppm)	1.47 * (0.92)	2.87 (1.71)	1.67 ^#^ (0.87)	<0.001
	1.21 * [0.31, 5.69]	2.47 [0.39, 13.8]	1.48 ^#^ [0.53, 7.54]	
CO exposure during CRECER (ppm)	0.89 * (0.72)	0.89 * (0.53)	1.30 (0.55)	<0.001
	0.67 * [0.31, 5.69]	0.75 * [0.30, 4.32]	1.18 [0.55, 4.13]	
Cumulative CO before allergy questionnaire ^ (ppm-year)	4.16 (2.69)	6.21 (3.19)	5.12 (3.07)	<0.001
	3.43 [0.88, 26.1]	5.50 [1.63, 24.0]	4.39 [1.40, 21.4]	

* Mean and median CO values during RESPIRE/CRECER labeled with * indicate measurements taken after chimney stove installation. ^#^ Group 3 index study children’s CO exposure during the RESPIRE study period was estimated based on their proxy infant siblings’ CO exposure during CRECER. ^ Cumulative CO exposures were calculated up to each child’s first allergic questionnaire, when children were on average 3.6 years old.

**Table 3 ijerph-19-14064-t003:** Skin Prick Test (SPT) results and self-reported allergic symptoms by study group.

	Group 1 (Baseline)	Group 2	Group 3	*p* Value for Equal Proportions
*Plancha* Installation Age	(<6 Months)	(~18 Months)	(~57 Months)	
	Number of Cases (%)	Number of Cases (%)	Number of Cases (%)	
**Positive SPT result**	n = 184	n = 187	n = 155	
*D. farinae*	29 (15.8%)	25 (13.4%)	17 (11.0%)	0.44
*D. pteronyssinus*	9 (4.9%)	11 (5.9%)	9 (5.8%)	0.90
Cockroach	70 (38.0%)	69 (36.9%)	56 (36.1%)	0.93
Dog	1 (0.5%)	6 (3.2%)	2 (1.3%)	0.13
Cat	5 (2.7%)	4 (2.1%)	3 (1.9%)	0.88
Ragweed	1 (0.5%)	1 (0.5%)	0 (0%)	0.66
**Maternal-reported allergic symptoms**	n = 188	n = 192	n = 157	
**Asthma symptoms**				
Any wheezing episode	13 (6.9%)	12 (6.2%)	29 (18.5%)	<0.001
Exercise-induced wheezing	8 (4.3%)	6 (3.1%)	9 (5.7%)	0.49
Cough	54 (28.7%)	40 (20.8%)	60 (38.2%)	<0.01
**Rhinitis symptoms**				
Sneezing, congestion, or rhinorrhea	33 (17.6%)	47 (24.5%)	54 (34.4%)	<0.01
**Allergic conjunctivitis symptoms**				
Itchy or watery eyes	33 (17.6%)	26 (13.5%)	27 (17.2%)	0.21
**Eczema symptoms**	24 (12.8%)	25 (13.0%)	22 (14.0%)	0.94

Note: n indicates the number of children in each group with valid SPT results or maternal-reported allergic symptoms in each group.

**Table 4 ijerph-19-14064-t004:** Odds Ratios (ORs) and 95% confidence intervals allergic outcomes associated with study groups.

	Group 2			Group 3		
*Plancha* Installation Age	(~18 Months)			(~57 Months)		
	Model 1	Model 2	Model 3	Model 1	Model 2	Model 3
Positive SPT result						
*D. farina*	0.82 (0.46, 1.47)	0.77 (0.41, 1.42)	0.81 (0.43, 1.53)	0.66 (0.34, 1.24)	0.72 (0.36, 1.44)	0.72 (0.35, 1.46)
*D. pteronyssinus*	1.22 (0.49, 3.08)	1.31 (0.50, 3.57)	1.24 (0.45, 3.51)	1.20 (0.46, 3.15)	1.78 (0.61, 5.26)	1.67 (0.54, 5.21)
Cockroach	0.95 (0.63, 1.45)	0.93 (0.60, 1.45)	0.93 (0.60, 1.46)	0.92 (0.59. 1.43)	0.91 (0.56, 1.46)	0.87 (0.53, 1.41)
Dog	6.07 (1.02, 115.17)	8.11 (1.07, 178.7)	11.25 (1.19, 437.76)	2.39 (0.23, 51.77)	2.96 (0.18, 82.70)	2.54 (0.10, 133.51)
Cat	0.78 (0.19, 3.00)	0.49 (0.09, 2.30)	0.47 (0.08, 2.25)	0.71 (0.14, 2.93)	0.45 (0.07, 2.29)	0.40 (0.06, 2.11)
Asthma symptoms						
Any wheezing episode	0.90 (0.39, 2.03)	0.74 (0.30, 1.81)	0.74 (0.30, 1.82)	3.05 (1.56, 6.28)	2.44 (1.14, 5.47)	2.42 (1.11, 5.48)
Exercise-induced wheezing	0.73 (0.23, 2.13)	0.59 (0.16, 2.03)	0.54 (0.15, 1.91)	1.37 (0.51, 3.73)	0.83 (0.25, 2.73)	0.72 (0.21, 2.42)
Cough	0.65 (0.41, 1.04)	0.65 (0.39, 1.08)	0.67 (0.40, 1.11)	1.53 (0.98, 2.41)	1.42 (0.86, 2.34)	1.60 (0.96, 2.67)
Rhinitis symptoms						
Sneezing, congestion, or rhinorrhea	1.52 (0.93, 2.52)	1.47 (0.86, 2.57)	1.47 (0.85, 2.67)	2.46 (1.50, 4.09)	2.06 (0.18, 3.63)	2.01 (1.13, 3.58)
Allergic conjunctivitis symptoms						
Itchy or watery eyes	1.32 (0.71, 2.47)	1.04 (0.52, 2.08)	1.01 (0.50, 2.03)	1.74 (0.94, 3.28)	1.41 (0.71, 2.85)	1.30 (0.64, 2.66)
Eczema symptoms	1.02 (0.56, 1.87)	1.04 (0.54, 2.02)	1.05 (0.54, 2.03)	1.11 (0.59, 2.08)	0.86 (0.42, 1.72)	0.82 (0.40, 1.67)

Note: Model 1 is unadjusted; model 2 adjusted for age, sex, second-hand smoke, number of children in the family, average weekly *temazcal* use time of the child, kitchen location and type, maternal asthma, maternal rhinitis, maternal eczema, numbers and species of animals owned; model 3 adjusted for maternal and paternal education, SES (land-owning, house-owning and number of major assets) in addition to covariates in model 2.

**Table 5 ijerph-19-14064-t005:** Odds Ratios (OR) and 95% confidence intervals (CIs) of allergic outcomes associated with cumulative CO exposure (ppm-year).

	Model 1	Model 2	Model 3
Positive SPT result			
*D. farina*	0.98 (0.90, 1.06)	0.96 (0.87, 1.05)	0.96 (0.86, 1.05)
*D. pteronyssinus*	0.95 (0.81, 1.08)	0.98 (0.83, 1.12)	0.99 (0.83, 1.12)
Cockroach	0.96 (0.90, 1.02)	0.97 (0.91, 1.04)	0.97 (0.91, 1.04)
Dog	1.01 (0.78, 1.19)	1.02 (0.76, 1.25)	1.04 (0.77, 1.30)
Cat	1.00 (0.80, 1.16)	1.00 (0.77, 1.22)	1.00 (0.76, 1.22)
Asthma symptoms			
Any wheezing episode	0.94 (0.84, 1.04)	0.98 (087, 1.10)	0.99 (0.87, 1.10)
Exercise-induced wheezing	0.99 (0.85, 1.12)	1.07 (0.90, 1.23)	1.06 (0.89, 1.22)
Cough	0.96 (0.90, 1.02)	0.99 (0.92, 1.06)	0.99 (0.92, 1.07)
Rhinitis symptoms			
Sneezing, congestion, or rhinorrhea	1.05 (0.99, 1.11)	1.10 (1.02, 1.18)	1.09 (1.02, 1.18)
Allergic conjunctivitis symptoms			
Itchy or watery eyes	1.07 (0.99, 1.14)	1.12 (1.03, 1.22)	1.11 (1.02, 1.21)
Eczema symptoms	0.98 (0.89, 1.06)	1.02 (0.92, 1.11)	1.02 (0.92, 1.12)

Note: Model 1 is unadjusted; model 2 adjusted for age, sex, second-hand smoke, number of children in the family, average weekly temascal use time of the child, kitchen location and type, maternal asthma, maternal rhinitis, maternal eczema, numbers and species of animals owned; model 3 adjusted for maternal and paternal education, SES (land-owning, house-owning and number of major assets) in addition to covariates in model 2. Number of children included in the analysis is 525 for all models.

## Data Availability

The data presented in this study are available on request from the corresponding author. The data are not publicly available due to privacy concerns.

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
