# Peer review of "Biomass Smoke Exposure and Atopy among Young Children in the Western Highlands of Guatemala: A Prospective Cohort Study"

_ijerph, 2022, doi:10.3390/ijerph192114064_

Round 1

Reviewer 1 Report

General comments

The paper by Lu et al. presents a well-designed prospective cohort study to examine the relationship between household air pollution, aeroallergen sensitization, and atopy in pediatric populations. Generally speaking, this paper is well organized and have high health relevance thus lies in the scope of IJERPH. My recommendation is accept, with minor revisions noted

Specific comments

Materials and Methods:

Line 108-113: Table 1 should be cited here.

Line 118-121: I understand why group 3 index study children did not have CO measurements obtained when they were <18 months of age. However, a discussion of the uncertainties associated with the use of personal CO exposures of their younger infant siblings as a proxy is needed.

Line 158-161: As shown in Fig.1 and Table 2, group 2 had higher CO exposure level than group 3. So whether it was appropriate to regard group 2 and group as intermediate and highest levels of biomass smoke exposure, respectively.

Table 2: Why CRECER proxy child CO (ppm) belongs to group 3? As mentioned by the authors,  new CRECER households (group 3) received the stove when the index children were approximately 5 years old, and their proxy infant siblings were 18 – 24 months old.

Discussion:

I would like to see a Table to compare the results of this study to those of prior studies.

Reviewer 2 Report

The authors investigated whether vented chimney stove can reduce the children’s HAP exposure and has lower the risks of allergic sensitization. This manuscript is well written. I have only two comments, as shown below.

1.  The presentation of the tables 2 and 3 is not easy to read, please modify them.

2. In line 117, the CO concentration as a proxy to represent the exposure of participants from cooking activity based on Refs. 20 and 21. However, previous studies directly measured the CO concentration for old chimney stove or open fire during the periods of cooking activity, not for new chimney stove. My question is whether the CO concentration is still a good proxy in study houses with new chimney stove?

Reviewer 3 Report

I think the results of the study are insipid because there have been some epidemiological studies on the relationship between solid fuel smoke and asthma and rhinitis in children. Of course, I can't deny that your investigation of allergens and allergic symptoms of rural children is meaningful and your research design is reasonable. I mean that you need to review more on existing relevant studies, reflecting the difference from this study. In addition, I have some minor comments.

1. You need to explain why the factors of ambient pollution are not considered in the model because the possibility of outdoor exposure increases when chimney stove replace open fire. And the average duration of cooking should also be an important confounding factor.

2. Group 2 and 3 are intermediate and high levels of biomass smoke exposure, but the results in Table 2 seem to be opposite. How to explain?

3. In line 159, you mentioned that cumulative CO exposure was used as a linear continuous exposure variable, but I don’t find that this variable was used in the analysis.
